# More Enhanced Swing Colpitts Oscillators: A Circuit Analysis

**Tatsuya Nomura and Toru Tanzawa ***

Graduate School of Integrated Science and Technology, Shizuoka University, Hamamatsu 432-8561, Japan
* Correspondence: toru.tanzawa@shizuoka.ac.jp

**Abstract:** In this paper, we show that an additional inductor–capacitor–inductor filter can increase the oscillation amplitude of the enhanced swing Colpitts oscillator (ESCO), and call this topology the more enhanced swing Colpitts oscillator (mESCO). When it is connected with a rectifier, the DC–DC boost conversion ratio can be increased, especially for low-voltage sensor ICs or energy harvesting. This paper focuses on the electrical characteristics of mESCO. The oscillation frequency was modeled as a function of the circuit parameters of mESCO. The common gate voltage gain ($A_{CG}$), defined by the ratio of the drain voltage amplitude to the source voltage amplitude of the switching MOSFET of mESCO, was also modeled under the assumption that all the circuit elements are ideal. The model was validated with a SPICE simulation. For $A_{CG} < 1.5$, the model was in good agreement with the SPICE results within 10%. In addition, the drain voltage amplitude $v_{da}$ was modeled by assuming that the average transconductance of the MOSFET in a half cycle is null when the long-channel Shockley model is used. $v_{da}$ needs to be sufficiently high to have a large DC–DC boost conversion ratio. The model can predict the tendency that $v_{da}$ increases as $A_{CG}$ approaches unity. We found that the voltage difference of the drain voltage amplitude to the source voltage amplitude is a constant even when the circuit parameters, and thereby $A_{CG}$ are varied.

**Keywords:** enhanced swing Colpitts oscillator; circuit analysis; circuit model

## 1. Introduction

Battery-free IoT sensor modules are required to eliminate battery replacement to reduce costs. Wireless power transfer (WPT) charges the modules with electromagnetic waves [1–4]. An energy transducer (ET) transforms environmental energy such as lights, heat flow, and vibration into electric power [5,6]. Those techniques can remove the batteries out of the modules. Circuit designs have improved the power efficiency of power converters even with low power received via WPT and ET. Multisine WPT can increase the peak power with the same average transmitter power, resulting in improvement in power efficiency [7–9]. Figure 1 graphically explains how it works. Each of the N multisine waves has power of 1/N to attain the same power as the conventional continuous wave. At the receiving antenna, the peak voltage amplitude can be theoretically increased by a factor of $\sqrt{N}$ with beating multisine waves when the input impedance is common for N multisine waves with different frequencies. Due to the nonlinearity of the rectifying diode, the multisine waves can output more averaged current than the continuous wave.

The Colpitts oscillator is one of the oscillator circuit topologies that provides a carrier frequency for wireless communication [10–13]. The phase noise of the oscillator is a critical factor limiting the sensitivity of wireless communication. It is important to increase the output voltage amplitude of the oscillator to reduce the phase noise. In order to increase the voltage amplitude, an enhanced swing Colpitts oscillator (ESCO) was proposed [14–18]. What about adding beating to the ESCO to further increase the output voltage amplitude in DC–DC boost converter applications for low-voltage sensor ICs or energy harvesting? In this paper, we show that an additional inductor–capacitor–inductor filter can increase the oscillation amplitude of the ESCO, and call this topology the more enhanced swing Colpitts oscillator (mESCO). Rectifiers were connected with the output terminals of ESCO

and mESCO to output boosted voltages, called ER (ESCO followed by rectifier) and mER (mESCO followed by rectifier), respectively. A SPICE simulation was run to compare the performance of ER and mER. The result showed that mER output a higher open circuit voltage and higher output current at a certain voltage than ER.

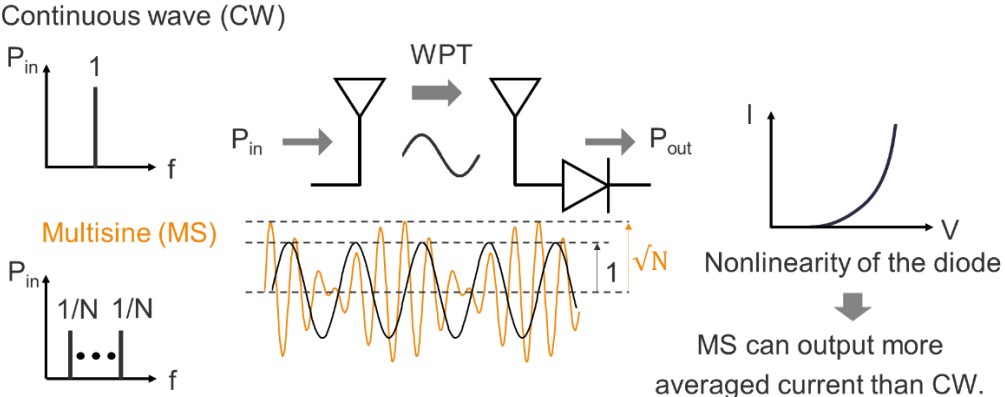

**Figure 1.** Efficiency improvement in wireless power transfer with multisine electromagnetic waves.

　　The rest of the paper discusses a circuit analysis of the mESCO. The oscillation frequency is shown as a function of the circuit parameters of mESCO. The common gate voltage gain $A_{CG}$, defined by the ratio of the drain voltage amplitude to the source voltage amplitude of the switching MOSFET of mESCO, was modeled under the assumption that all the circuit elements are ideal. The drain voltage amplitude $v_{da}$ was also modeled by assuming that an averaged transconductance of the MOSFET in a half cycle is null when the long-channel Shockley model is used. $v_{da}$ needs to be sufficiently high to have a large DC–DC boost conversion ratio. The model can predict the tendency that $v_{da}$ increases as $A_{CG}$ approaches unity. We also found that the voltage difference of the drain voltage amplitude to the source voltage amplitude is a constant even when the circuit parameters, and thereby $A_{CG}$, are varied. The paper is organized as follows: Section 2 proposes the mESCO and mER. The performance of mER is compared with that of ER. Section 3 analyzes the oscillation frequency, common gate voltage gain, and drain voltage amplitude of mESCO. The models are validated with SPICE simulation. The conclusion is given in Section 4.

## 2. More Enhanced Swing Colpitts Oscillator (mESCO), and DC–DC Converter with mESCO and Rectifier (mER)

　　Let us consider on-chip DC–DC boost converters composed of an oscillator whose peak output voltage is higher than the input DC voltage $V_{IN}$ and a rectifier to supply power to building blocks in sensor/RF IC, as shown in Figure 2. When the decoupling capacitor $C_{DEC}$ is too large to stabilize the input power rails, the oscillator runs with the DC supply voltage, as shown in Figure 2a. On the other hand, if $C_{DEC}$ is set to a relatively small value, the AC component adds to the DC supply voltage to create beat tones into the oscillator, as shown in Figure 2b.

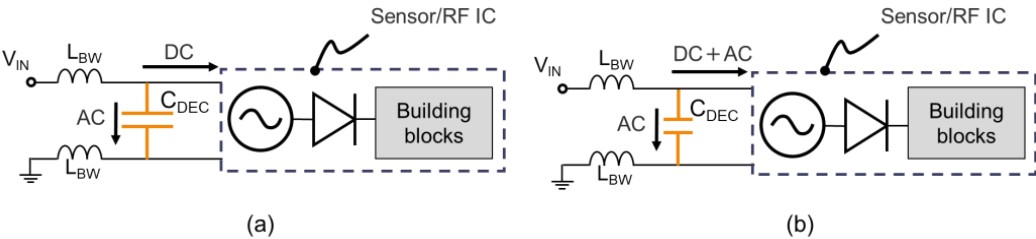

**Figure 2.** (**a**) Relatively large decoupling capacitor to filter out the AC component; (**b**) relatively small decoupling capacitor to create beat tones into the oscillator.

Figure 3a illustrates a boost converter (ER) with an enhanced swing Colpitts oscillator (ESCO) followed by a rectifier to drive the load $R_L$. The drain voltage of M1 exceeds $V_{IN}$. M2 transfers the current when the peak drain voltage of M1 is higher than the target voltage $V_{OUT}$. To increase the $V_{SS}$ of the ESCO, a small decoupling capacitor is connected between the $V_{IN}$ and $V_{SS}$ through the parasitic inductance of bonding wires, as shown in Figure 3b. Additional oscillation at $V_{SS}$ increases the drain voltage of M1, resulting in a higher output current at the same $V_{OUT}$ as ER.

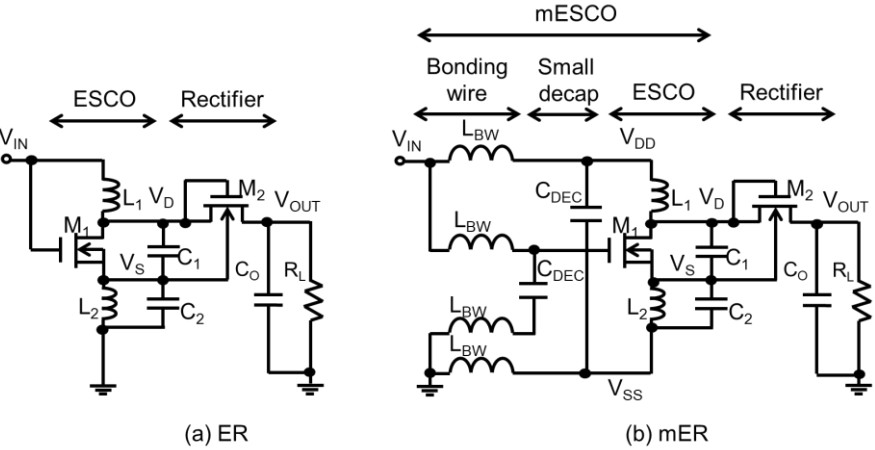

**Figure 3.** (**a**) Enhanced swing Colpitts oscillator (ESCO) followed by rectifier (ER); (**b**) ER with LC filter (more enhanced swing Colpitts oscillator followed by rectifier (mER).

A SPICE simulation was run with $V_{IN} = 0.3$ V, $L_{BW} = 2$ nH, $C_{DEC} = 25$ pF, $L_1 = 0.239$ nH, $L_2 = 2.93$ nH, $C_1 = 4$ pF, and $C_2 = 1.5$ pF. The parameters $L_1$, $L_2$, $C_1$, and $C_2$ are common to both circuits. The SPICE model in 65 nm CMOS was used in this study. Figure 4a shows the waveform of $V_D$ and $V_S$ of ER and mER when the outputs are open. The highest peak voltages of $V_D$ were 1.08 V with ER and 2.01 V with mER. In mER, the peak voltages were alternately high and low. Figure 4b shows the $V_{OUT}$ vs. $I_{OUT}$ of ER and mER. In addition to an increase in the maximum attainable output voltage, mER can have a larger output current at any operating output voltage than ER. In Section 3, a circuit analysis for mESCO is conducted.

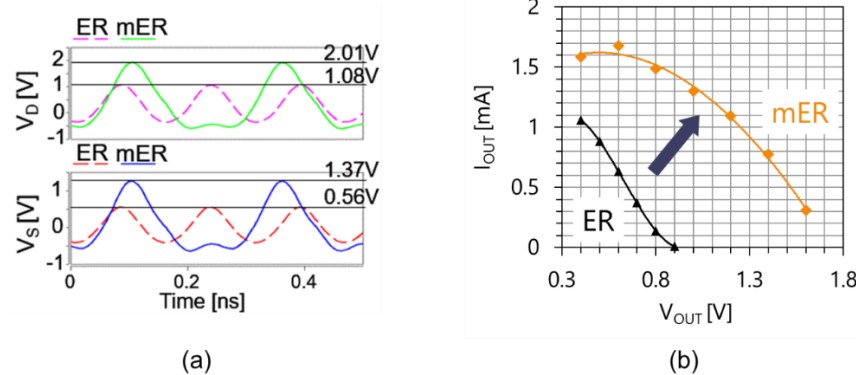

**Figure 4.** Comparison of mESCO with ESCO; (**a**) waveform of $V_D$ and $V_S$ when the outputs are open, (**b**) $V_{OUT} - I_{OUT}$.

## 3. Circuit Analysis of mESCO

Let us define the notation of voltages in this work, as shown in Figure 5. The DC offset and voltage amplitude are described by $V_X$ and $v_{xa}$, respectively. The voltage differences from the ground and DC offset are described by $V_x$ and $v_x$ as a function of phase $\theta$, respectively.

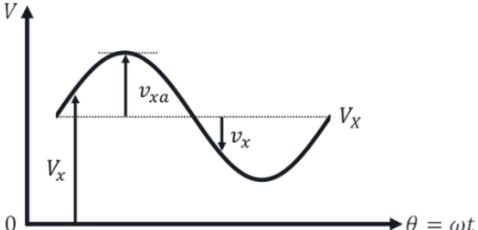

**Figure 5.** Notation of voltages in this work.

### 3.1. Oscillation Frequency

The oscillation frequency of ESCO was modeled in [16] by ignoring any real part of impedance and conductance, assuming the ESCO operates in steady state. We can similarly formulate the oscillation frequency of mESCO. Figure 6 shows the circuit reduction of mESCO from Figure 6a through Figure 6g. A small-signal equivalent circuit of Figure 6a is reduced to Figure 6b. Introducing $L_{tb}$ for the series connection of $L_t$ and $L_b$ with $L_{tb} = L_t + L_b$, and eliminating $M_1$ results in Figure 6c under the assumption that any real part of impedance and conductance is omitted for a simple model. When the impedances of the parallel connections of $L_{tb}$ and $C_{IN}$ and of $L_2$ and $C_2$ are capacitive, they can be replaced with single capacitors $C_{eq1}$ and $C_{eq2}$, respectively, by using the formula shown in Figure 6d, where $\omega$ is the angular velocity of a signal of interest. Likewise, when the impedance of the series connection of $L_1$ and $C_1$ is inductive, we can replace it with a single inductor $L_{eq}$, by using the formula shown in Figure 6e. Thus, the circuit shown in Figure 6c is reduced to Figure 6f using $L_{eq}$, $C_{eq1}$ and $C_{eq2}$. Finally, we can obtain an LC circuit as shown in Figure 6g, where $C_{eq} = C_{eq1} + C_{eq2}$. Thus, (1)–(4) hold for mESCO.

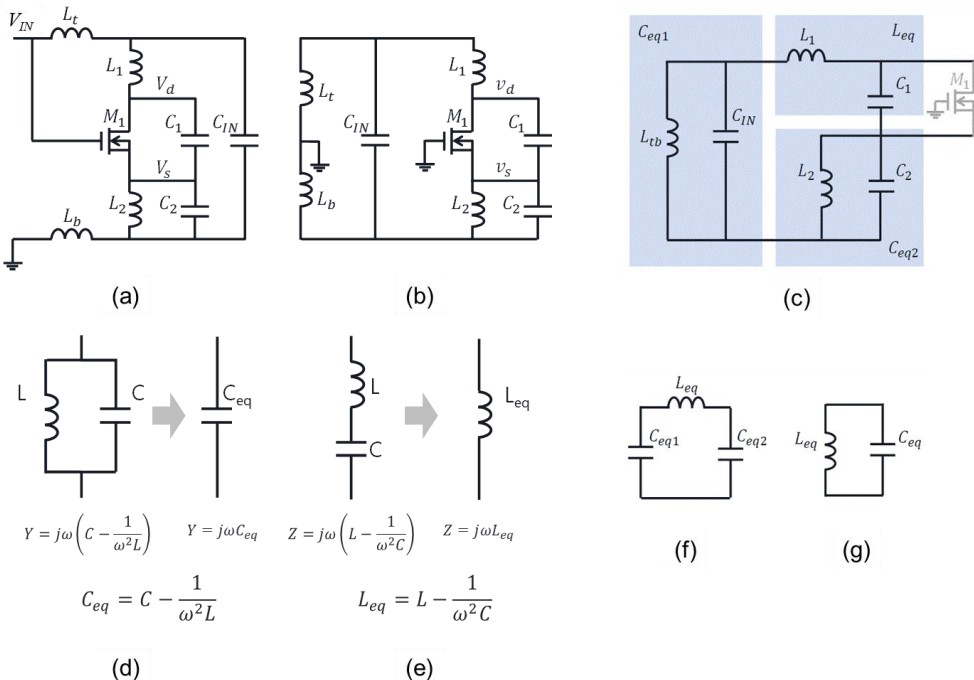

**Figure 6.** Circuit reduction of mESCO from (**a**) through (**g**); (**a**) original mESCO, (**b**) small-signal equivalent circuit of (**a**), (**c**) omitting M1, (**d**) equivalent capacitance of L and C connected in parallel, (**e**) equivalent inductance of L and C connected in series, (**f**) circuit reduced from (**c**) with (**d**) and (**e**), and (**g**) circuit reduced from (**f**).

$$C_{eq1} = C_{in} - \frac{1}{\omega^2 L_{tb}} \tag{1}$$

$$C_{eq2} = C_2 - \frac{1}{\omega^2 L_2} \tag{2}$$

$$L_{eq} = L_1 - \frac{1}{\omega^2 C_1} \tag{3}$$

$$C_{eq} = \frac{C_{eq1} C_{eq2}}{C_{eq1} + C_{eq2}} \tag{4}$$

$$f = \omega/2\pi = 1/2\pi \sqrt{L_{eq} C_{eq}} \tag{5}$$

A fundamental frequency is given by the solution in cubic Equation (5) in terms of $f^2$, and thereby can be resolved with three different values. To validate the models proposed in Section 3 with SPICE simulation, the nominal condition shown in Table 1 was used. The values of the inductors and capacitors were much larger than the ones fabricated in a single chip because the SPICE model in 180 nm CMOS was used for $M_1$. For parameter response on electrical characteristics, the remaining ones except for the sweeping parameter were set as shown in Table 1.

**Table 1.** Nominal condition to validate the models proposed in Section 3 with SPICE simulation.

| Parameter | Default Value |
|---|---|
| $L_1$ | 5 µH |
| $L_2$ | 16 µH |
| $L_t (=L_b)$ | 5 µH |
| $C_1$ | 3 nF |
| $C_2$ | 8 nF |
| $C_{IN}$ | 1 nF |
| $V_{IN}$ | 0.8 V |

The numerical solutions under wide ranges of circuit parameters were in good agreement with the SPICE results within 10% at most, as shown in Figure 7. Among the three solutions, the highest one was selected because the other two frequencies are too low to have $L_{eq}$ and $C_{eq}$ positive values.

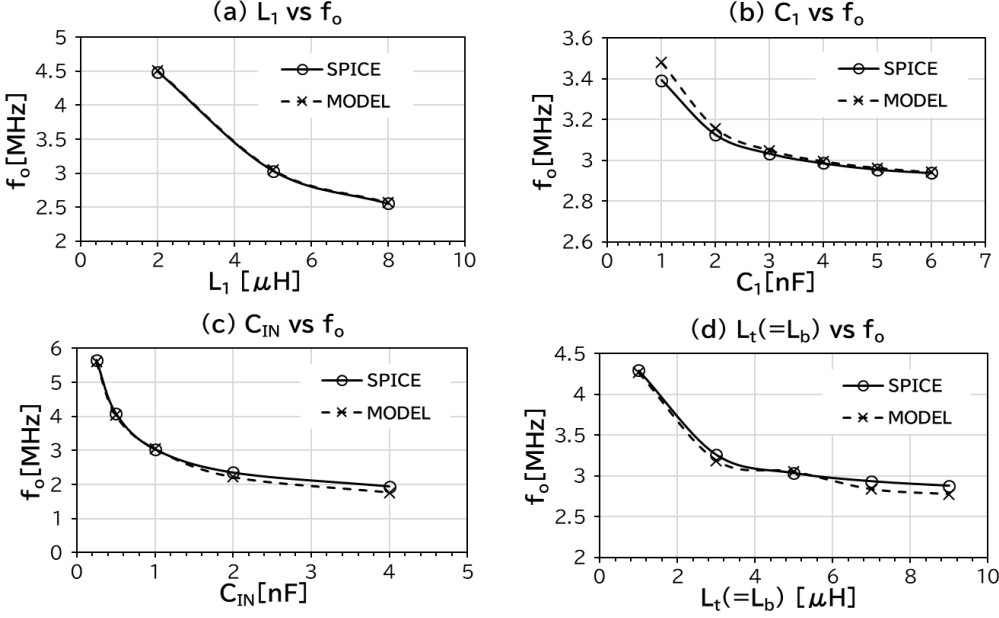

**Figure 7.** Operation frequency as a function of each circuit parameter. (**a**) $L_1$ vs. fo, (**b**) $C_1$ vs. fo, (**c**) $C_{IN}$ vs. fo, (**d**) $L_t(=L_b)$ vs. fo.

### 3.2. Common-Gate Voltage Gain

For DC–DC boost converters to have large boost ratios, a large voltage amplitude at the drain of *M1* is required. Let us determine a common gate voltage gain $A_{CG}$ as a function of the circuit parameters first. As performed for modeling the operation frequency of mESCO, a circuit transformation was produced from an original circuit diagram of mESCO as shown in Figure 8a to its small-signal equivalent circuit as shown in Figure 8b, and then a circuit model of mESCO as shown in Figure 8c.

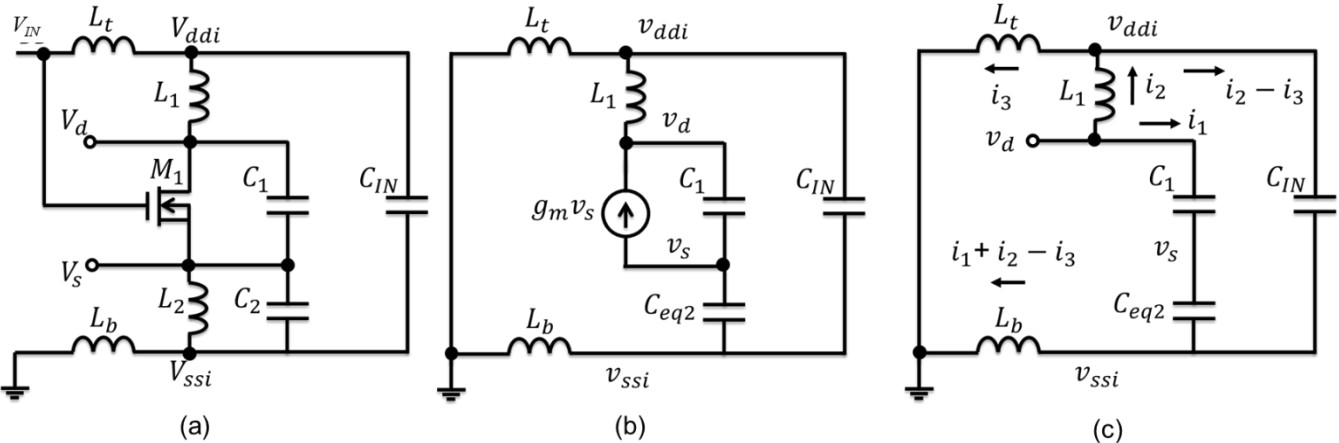

**Figure 8.** Circuit transformation: (**a**) an original circuit diagram of mESCO, (**b**) its small-signal equivalent circuit, and (**c**) a circuit model of mESCO to extract its common-gate voltage gain $A_{CG}$.

When $v_d$ is forced, what voltage appears at $v_s$? Seven variables (four voltages and three currents) among six equations were applied, as shown by (6)–(11).

$$v_{ssi} = j\omega L_b(i_1 + i_2 - i_3) \tag{6}$$

$$v_s - v_{ssi} = i_1/j\omega C_{eq2} \tag{7}$$

$$v_d - v_{ddi} = j\omega L_1 i_2 \tag{8}$$

$$v_{ddi} - v_{ssi} = (i_2 - i_3)/j\omega C_{IN} \tag{9}$$

$$v_{ddi} = j\omega L_t i_3 \tag{10}$$

$$v_d - v_s = i_1/j\omega C_1 \tag{11}$$

Therefore, an equation can relate any two variables. After some calculations, (12) was derived for the common gate voltage gain $A_{CG}$. With (13)–(18), (12) provides a value for $A_{CG}$ when the circuit parameters are given.

$$A_{CG} = v_d/v_s = T_{SQ1}/T_{SQ2} \tag{12}$$

$$T_{SQ1} = (1 + \beta)(T_{SQ3} - L_X C_X) + L_X C_{eq2} \tag{13}$$

$$T_{SQ2} = \beta(T_{SQ3} - L_X C_X) + L_t C_{IN} \tag{14}$$

$$T_{SQ3} = \omega^2 C_{IN}^2 L_1 L_t \tag{15}$$

$$\beta = C_1/C_{eq2} \tag{16}$$

$$L_X = \omega^2 C_{IN} L_1 L_t - L_1 - L_t \tag{17}$$

$$C_X = C_{eq2} + C_{IN} - 1/\left(\omega^2 L_b\right) \tag{18}$$

where $C_{eq2}$ is given by (2). $\beta$, $L_X$, and $C_X$, defined by (16), (17), and (18), respectively, were used to calculate $T_{SQ1-3}$. Figure 9 shows $A_{CG}$ as a function of each design parameter. The

tendencies of $A_{CG}$ vs. $L_1$ (a) and $C_1$ (b) are matched with those for ESCO [16]. As expected, the smaller the $C_{IN}$, the closer to unity for $A_{CG}$ in (c). (This tendency validated increasing the drain voltage amplitude with a smaller $C_{IN}$ in the following sub-section, as expected in Section 1). There were substantial discrepancies in $A_{CG}$ between the model and SPICE results for $C_{IN} > 2$ nF. A physical background is required here, but its investigation will be conducted in future work. Figure 9d suggests that *Lt* and *Lb* can be minor contributors to $A_{CG}$, resulting in no significant impact on the drain voltage amplitude, as discussed in the following subsection. In summary, the model calculation results were in good agreement with the SPICE results, except for $C_{IN} > 2$ nF.

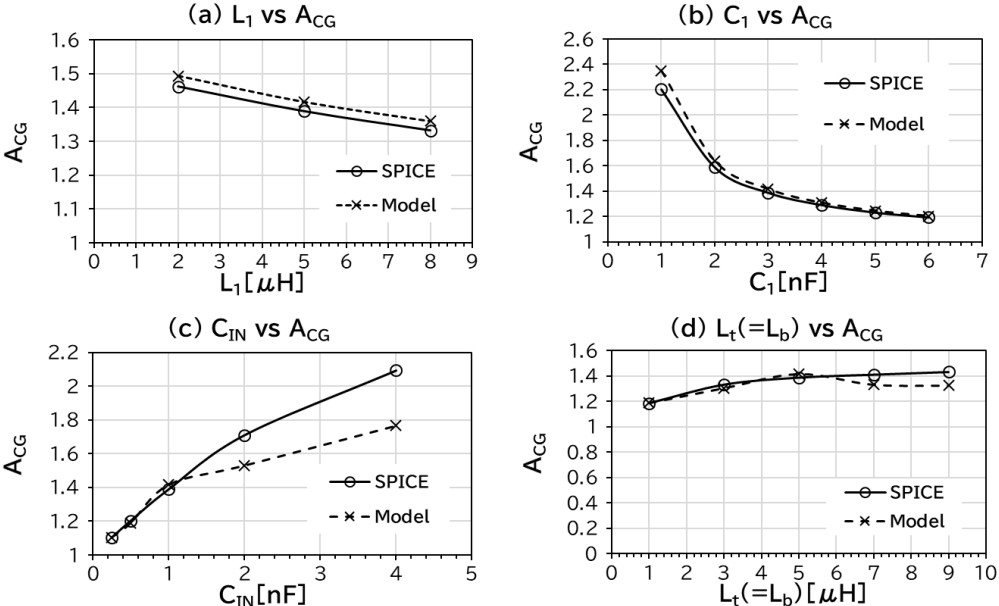

**Figure 9.** $A_{CG}$ as a function of each design parameter. (**a**) $L_1$ vs. $A_{CG}$, (**b**) $C_1$ vs. $A_{CG}$, (**c**) $C_{IN}$ vs. $A_{CG}$, (**d**) $L_t(=L_b)$ vs. $A_{CG}$.

### 3.3. Drain Voltage Amplitude $v_{da}$

In this subsection, the drain voltage amplitude $v_{da}$ is modeled. In reality, there was a distortion in $V_d$ and $V_s$ due to the nonlinear behavior of the transconductance and drain conductance of M1, but it was assumed for simplicity in this study that the AC components of $V_d$ and $V_s$ are modeled with sinusoidal waveforms whose amplitudes are $v_{da}$ and $v_{sa}$, respectively, as shown in Figure 10.

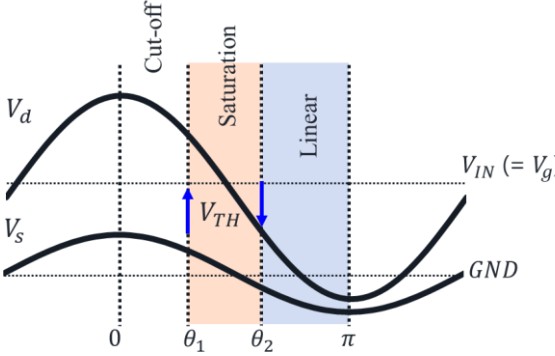

**Figure 10.** Waveform of $V_d$ and $V_s$ and operation modes in one cycle.

$V_d$ has an offset voltage of $V_{IN}$. Thus, $V_d$ and $V_s$ can be written as (19) and (20), respectively.

$$V_d = V_{IN} + v_{da} \cos \theta \tag{19}$$

$$V_s = v_{sa} \cos \theta = \frac{v_{da}}{A_{CG}} \cos \theta \tag{20}$$

When $v_{sa}$ is large enough, *M1* operates in three different modes, i.e., in cut-off, saturation, and linear operation modes. $\theta_1$ is the boundary between cut-off and saturation, resulting in (21), where $V_{TH}$ is the threshold voltage of *M1*. $\theta_2$ is the boundary between saturation and linear, resulting in (22).

$$\theta_1 = \cos^{-1} \frac{V_{IN} - V_{TH}}{v_{sa}} \tag{21}$$

$$\theta_2 = \cos^{-1} \frac{-V_{TH}}{v_{da}} \tag{22}$$

Let us assume a simple Shockley model for *M1*, as described in (23) and (24), where *Ids_sat* and *Ids_lin* are the drain current in saturation and liner modes, respectively, and *k* is a proportional coefficient of the drain current.

$$I_{ds\_sat} = \frac{1}{2} k \left( V_{gs} - V_{th} \right)^2 \tag{23}$$

$$I_{ds\_lin} = k \left\{ \left( V_{gs} - V_{th} \right) V_{ds} - \frac{1}{2} V_{ds}{}^2 \right\} \tag{24}$$

The transconductance is expressed by (25) and (26) in saturation and liner modes, respectively.

$$g_{m_{sat}} = \frac{\partial I_{ds_{sat}}}{\partial V_{gs}} = k \left( V_{gs} - V_{th} \right) \tag{25}$$

$$g_{m\_lin} = \frac{\partial I_{ds\_lin}}{\partial V_{gs}} = k V_{ds} \tag{26}$$

Let us assume (27) holds when mESCO runs in steady state, where the peak drain voltage does not change by cycle, because when the left side value of (27) is positive, the amplitude increases by a cycle; whereas when it is negative, the amplitude decreases by a cycle.

$$\int_0^\pi g_m(\theta) d\theta = 0 \tag{27}$$

Using (21), (22), (25) and (26), (27) resulted in (28).

$$\begin{aligned} \int_0^\pi g_m(\theta) d\theta &= \int_{\theta_1}^{\theta_2} g_{m\_sat}(\theta) d\theta + \int_{\theta_2}^\pi g_{m\_lin}(\theta) d\theta \\ &= V_{IN}(\pi - \theta_1) - V_{th}(\theta_2 - \theta_1) + v_{sa} \sin \theta_1 - v_{da} \sin \theta_2 = 0 \end{aligned} \tag{28}$$

Equation (28) shows that $v_{da}$ is a function of $A_{CG}$, $V_{IN}$ and $V_{TH}$. Therefore, the model was compared with the SPICE results in terms of $v_{da}$ vs. $A_{CG}$, as shown in Figure 11a. Dots in colors show the SPICE results where the circuit parameters are varied as much as the ones in Figure 9. Because the SPICE model for NMOSFET used in this study has no breakdown at high drain voltages, all the simulated data were plotted. However, because there is a strict specification for the maximum voltages in reality, it can limit the design space for the circuit parameters. Even though the model result has an offset from the trend curve of the SPICE results, the model well-represents the tendency that $v_{da}$ is determined by $A_{CG}$ rather than individual circuit parameters. It is interesting to note that the data points of $v_{da}$ and $v_{sa}$ are plotted on a linear line with a slope of 1.00 for both the SPICE results and model calculated results with a slightly different offset of 1.05 and 1.25 V for the SPICE results and model calculated results, respectively, as shown in Figure 11b.

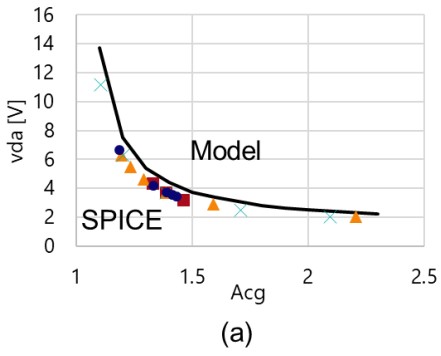 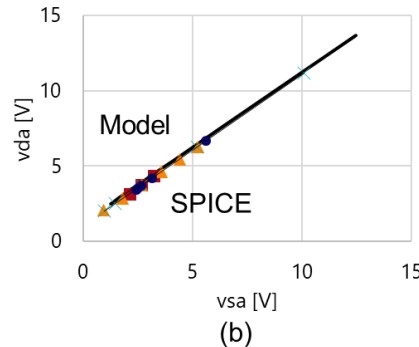

(a)

(b)

**Figure 11.** (**a**) $A_{CG}$ vs. $v_{da}$ and (**b**) $v_{sa}$ vs. $v_{da}$.

Figure 12a,b show the waveforms for voltage, current, and *gm* in cases of $A_{CG}$ = 1.2 and 1.8, respectively. The waveform for *gm* in Figure 12a suggests that *gm* in a linear operation can partially contribute to surplus for a very large voltage amplitude. For a relatively low voltage swing in Figure 12b, positive and negative *gm* appear in the saturation and linear regions, respectively.

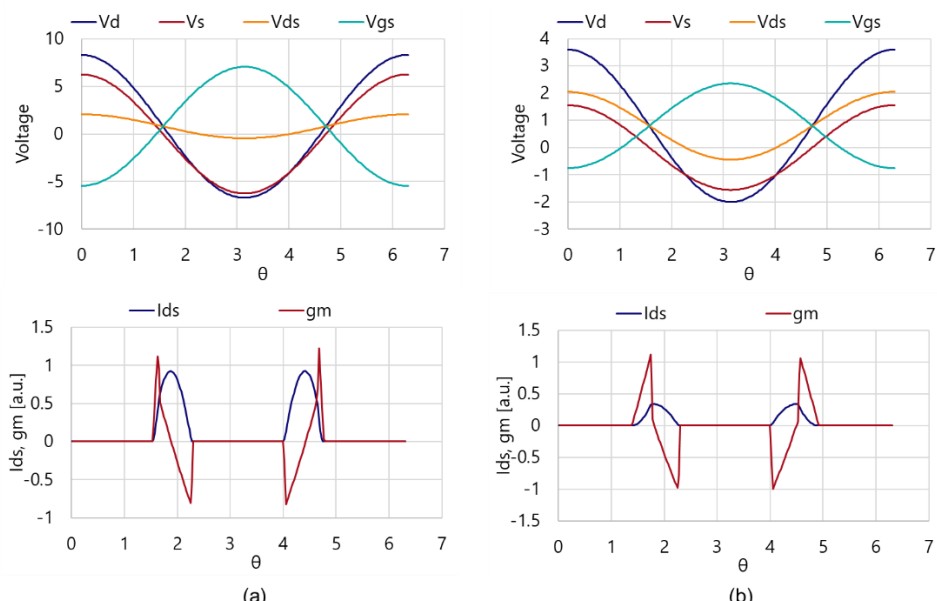

(a)

(b)

**Figure 12.** Waveform in cases of $A_{CG}$ = 1.2 (**a**) and 1.8 (**b**).

To compare the $v_{da}$ of mESCO with that of ESCO, SPICE simulations and model calculations based on [16] were made with the same parameter conditions for $L_1$, $C_1$, and $V_{IN}$ as mESCO. Figure 13 shows the comparison results. Even though the model accuracy of (28) for mESCO is not as sufficient as that of [16] for ESCO, the trend over circuit parameters represents the SPICE results. With the same values for $L_1$, $L_2$, $C_1$, and $C_2$ as those of ESCO, mESCO has a 2× or larger $v_{da}$ than ESCO.

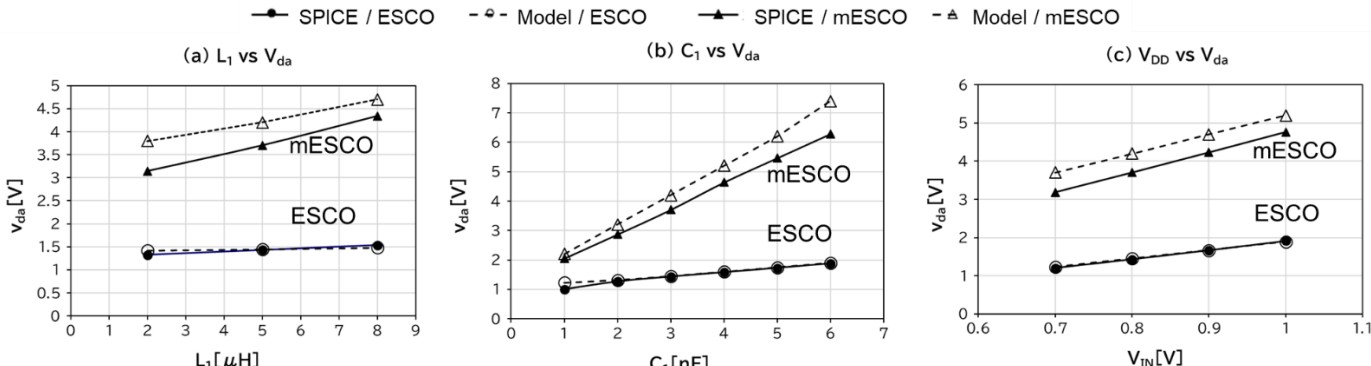

**Figure 13.** $v_{da}$ of ESCO and mESCO as a function of $L_1$ (**a**), $C_1$ (**b**), and $V_{IN}$ (**c**).

The data in Figure 13 are placed in a single $v_{da}-A_{CG}$ graph as shown in Figure 14. mESCO, as well as ESCO, has a unified characteristic curve. It is essential to design mESCO and ESCO with $A_{CG}$ as close to unity as possible to have a large $v_{da}$.

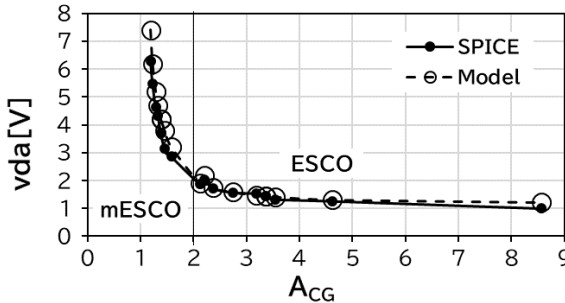

**Figure 14.** $v_{da}$ of ESCO and mESCO as a function of $A_{CG}$.

## 4. Conclusions

We found that an additional L–C–L filter for ESCO, called mESCO, increased the peak drain voltage, which will contribute to increasing the boost ratio when it is used in DC–DC boost converter applications. In this study, the operation frequency, common gate voltage gain, and drain voltage amplitude were analyzed with simple proposed models for mESCO. Even though the frequency model potentially has three different frequencies as a solution, the lower two components were not allowed because an equivalent inductor and capacitor would have had negative inductance and capacitance, respectively. The highest frequency was matched with the SPICE results in wide circuit parameter ranges. The common gate voltage gain was also modeled with a simple circuit transformation. The model calculation results are in good agreement with the SPICE results within a 10% error, except for large $C_{IN}$. In addition, the drain voltage amplitude was modeled with an assumption that the average transconductance of the switching transistor over a half cycle is null. Even with a simple Shockley model, the drain voltage amplitude was successfully modeled so that it is a function of the common gate voltage gain rather than individual circuit parameters. This fact is valid for ESCO as well as mESCO. In wide ranges of $L_1$, $C_1$, and $V_{IN}$, the drain voltage amplitude of mESCO was 2× or larger than that of ESCO. ESCO and mESCO had the same trend in the drain voltage amplitude over the common gate voltage gain.

Further experiments to validate the model and SPICE results and an application of mESCO to DC–DC boost converters will be conducted in the future. When a rectifier is added to mESCO for the DC–DC boost converter, it would have another impedance at the drain of the switching transistor, potentially yielding a loss in the drain voltage amplitude and thereby a degradation of the output power of the DC–DC converter. One would need to analyze its impact to design a DC–DC converter with sufficient output power.

**Author Contributions:** Conceptualization, T.T.; methodology, T.N. and T.T.; software, T.N.; validation, T.N. and T.T.; formal analysis, T.N. and T.T.; investigation, T.N. and T.T.; writing—original draft preparation, T.N.; writing—review and editing, T.T.; funding acquisition, T.T. All authors have read and agreed to the published version of the manuscript.

**Funding:** This research was funded by JSPS KAKENHI grant number 22K04042.

**Institutional Review Board Statement:** Not applicable.

**Informed Consent Statement:** Not applicable.

**Data Availability Statement:** Not applicable.

**Acknowledgments:** This work was supported by JSPS KAKENHI grant number 22K04042.

**Conflicts of Interest:** The authors declare no conflict of interest.

## Nomenclature

| | |
|---|---|
| $A_{CG}$ | Common gate voltage gain |
| $C_{DEC}$ | Decoupling capacitor |
| ER | ESCO followed by rectifier |
| ESCO | Enhanced swing Colpitts oscillator |
| ET | Energy transducer |
| IoT | Internet of Things |
| $L_{BW}$ | Inductance of bonding wires |
| mER | mESCO followed by Rectifier |
| mESCO | more Enhanced Swing Colpitts Oscillator |
| SPICE | Simulation Program with Integrated Circuit Emphasis |
| $v_{da}$ | Drain voltage amplitude |
| $V_{IN}$ | Input DC voltage |
| $V_{OUT}$ | Output DC voltage |
| $V_X$ | DC offset |
| $V_x$ | Voltage difference from ground |
| $v_x$ | Voltage difference from the DC offset |
| $V_{xa}$ | Voltage amplitude |
| WPT | Wireless power transfer |
| $\theta$ | Phase |

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
