# Peer review of "More Enhanced Swing Colpitts Oscillators: A Circuit Analysis"

_electronics, doi:10.3390/electronics11182808_

Round 1

Reviewer 1 Report

The paper deals with the analysis of the electrical characteristics of a variant of the Enhanced Swing Colpitts Oscillator (ESCO) circuit to which a filter is added at the input. Its electrical behaviour is studied. The research approach is rigorous and clear in its conclusions. The research contribution is relevant and the application in "Electronics" is clear. In the reviewer's opinion, some minor modifications need to be considered:

- Section with abbreviations and acronyms used in the text should be added. In some cases the acronym is not properly identified (e.g. IoT, BSIM, LHS, etc).

- The authors should justify the use of PSPICE as a simulation tool.

- The reduction of the equivalent circuit from (a) to (b) in Figure 6 is not well explained.

- The parameters of the equations (e.g. β, ω, k) should be adequately described.

Author Response

Dear the reviewer,

The authors wish to thank you for providing valuable comments and suggestions. Each has been responded in a revised version of the manuscript. Can you please check if it has been correctly done? 

Best regards,
Toru Tanzawa 

Reviewer 2 Report

The manuscript successfully demonstrates how the addition of an inductor-capacitor-inductor filter increases the peak drain voltage in an Enhanced Swing Colpitts Oscillator, additionally increasing the boost ratio when utilizing DC-DC boost converters. Results are validated via a SPICE simulation and compared to a rectified Enhanced Swing Colpitts Oscillator.

The scientific approach is sound and the paper is reasonably organized. I do have a few suggestions for improvement:

-Future work is only very briefly mentioned (often in the context of a single sentence). It would be helpful for these ideas to be expanded upon.

-The figures would benefit from more thorough captions.

-It would be informative to provide more details regarding the calculations mentioned in “Therefore, one can have an equation to relate any two variables. After some calculations, one can have the following equations”

-Is there any intuition for the discrepancies in ACG between model and SPICE results for large C_{in}?

- There are some minor grammatical issues and typos throughout that should be carefully addressed, often regarding articles (the, a, an)

Ex: “Colpitts oscillator is one of the oscillator…” -> “The Colpitts oscillator is one of the oscillator…”

Author Response

(The authors gave the same response as above.)

Reviewer 3 Report

The work done by the authors is satisfactory, paper writing and presentation is good. However, conclusion should be modified. It is almost identical to the abstract. Various comparative analysis of ESCO and mESCO have been presented, this should be used to modify the conclusion section.

Author Response

(The authors gave the same response as above.)
